plant science/ecology/environmental science

bankrupt bush, *Stoebe vulgaris*, plant species diversity, conservation, encroachment, Bankenveld

**Author for correspondence:**
Susannah C. Graham
e-mail: patrosc@unisa.ac.za

# Impact of *Seriphium plumosum* densification on Mesic Highveld Grassland biodiversity in South Africa

Susannah C. Graham, Alan S. Barrett and Leslie R. Brown

Applied Behavioural Ecology and Ecosystem Research Unit, Department of Nature Conservation, UNISA, Private Bag X6, Florida 1710 Republic of South Africa

SCG, 0000-0001-9615-8432

Mesic Highveld Grassland is important for biodiversity conservation, but is threatened by bush densification from *Seriphium plumosum*. This indigenous densifier spreads rapidly and outcompetes other herbaceous species, changing the species composition and structure of grasslands. This study looks at three different densities of *S. plumosum* and how these affect grassland biodiversity within Telperion, Mpumalanga, South Africa. An intermediate density of *S. plumosum* (1500 individuals (ind) ha$^{-1}$) resulted in the highest plant species diversity ($H = 2.26$), a low density (24 ind ha$^{-1}$) was moderately diverse ($H = 1.96$) and a high density (9500 ind ha$^{-1}$) was least diverse ($H = 1.78$). There were differences between the three densities in terms of plant species diversity, with the intermediate density being significantly more diverse ($p < 0.01$) than both the low and high densities. Findings indicate that there was a significant difference between the sites in terms of ecological successional status ($p < 0.01$). The presence of *S. plumosum* at low densities can be considered an integral part of the environment. It is important that in areas where *S. plumosum* occurs, it should be monitored. If this species is not in balance within its environment and it starts becoming dense, it will negatively affect the biodiversity, species composition and structure of the habitat.

## 1. Introduction

The Grassland biome of South Africa covers 29% of the country and occurs in eight provinces [1]. This biome is the second largest biome in South Africa and has the second highest biodiversity [2]. Grasslands are complex and slowly evolved ecosystems with a high biodiversity [2]. Biodiversity is defined as the number of different species present in an area and refers

to how their abundance is spread within an ecological community [3]. It is considered that the higher the diversity, the more resilient an ecosystem will be to disturbance. Tilman & Downing [4] found that greater plant species richness resulted in greater ecosystem stability. Grasslands provide habitats for many rare, endangered and endemic plant and animal species [2]. Hilton-Taylor [5] confirmed that of the 640 Red Data listed species found in grasslands, 136 are endangered.

Large sections of the Grassland biome of South Africa have been transformed due to agriculture, forestry, urbanization and mining [6] that has led to a loss of biodiversity in these sensitive ecosystems. It is estimated that more than 40% of the Grassland biome has been permanently modified. The remaining 60% is mostly classified as threatened, while less than 3% is under formal protection [2]. Another threat to grassland biodiversity is the densification/encroachment by woody species. Bush densification/encroachment has serious consequences for biodiversity [7] and is also linked to a decrease in ecosystem functions and processes [8–10], which are enhanced by incorrect land management practices such as overgrazing. The impacts of climate change encourage the growth and establishment of woody species within grasslands due to an increase in temperatures and a decline in the number of frost events. The regular occurrence of frost in grasslands assists in controlling woody species by killing seedlings and suppressing growth of larger trees and shrubs. It is predicted that the increased survival of woody species seedlings will lead to bush encroachment and the conversion of grasslands into savannah ecosystems [11].

The dwarf shrub *Seriphium plumosum* (colloquially known as bankrupt bush) is a woody species [12] that encroaches/densifies in grasslands causing veld condition degradation with an associated loss in production. This indigenous dwarf shrub is found primarily in the Fynbos and Grassland biomes [13]. It establishes in poorly managed veld as well as in abandoned cultivated areas [14]. In these areas, it outcompetes local herbaceous species and tends to dominate, changing the species composition and structure of the vegetation. *Seriphium plumosum* has an average height and width of 60 cm. It is well adapted to survive harsh conditions, with a light grey colour that reflects sunlight, a woolly indumentum for protection and reduced leaf size to assist in decreasing water loss. Its root system can reach a diameter of 1 m$^3$ and penetrates up to 1.8 m into the ground [14]. It has been noted by Snyman [15] that *S. plumosum* prefers rocky, infertile, sandy soils. The plant has aromatic, volatile oils, which makes it unpalatable [15]. *Seriphium plumosum* suppresses grass growth due to its shading effect, change in soil moisture and allelopathic properties which inhibit seed germination and growth of other species [16]. It is destroying plant biodiversity and has a knock-on effect for grassland animal species [12].

Fire is essential in grasslands to retain vegetation structure and manage woody biomass [17]. However, it is argued that the use of fire to control *S. plumosum* may increase germination thereof and does not kill adult plants [14,18]. A number of control methods are presently used in an attempt to eradicate *S. plumosum* from grasslands, including selective clearing, manual removal and chemical treatment with herbicide. Most control methods are time-consuming and labour-intensive, especially on a large scale [12]. If not properly 'managed', this species could deprive existing plants of available resources, forming a monoculture of mature plants that are unpalatable [15].

This study investigates how different densities of *S. plumosum* affect the biodiversity of a mesic grassland. The objective was to understand how grassland plant species composition, biodiversity and community structure were affected by various densities of *S. plumosum*.

# 2. Methods

## 2.1. Study area

Telperion Nature Reserve (Telperion) is located approximately 25 km east of Bronkhorstspruit and 45 km west of eMalahleni (latitude 25°41'35.20" S and longitude 29°0'7.01" E) in the Mpumalanga Province of South Africa. It is approximately 9000 ha in size and is situated on the eastern extremity of the Magaliesberg Mountain Range, east of Pretoria (figure 1). The reserve falls within the Rand Highveld Grassland which forms part of the Mesic Highveld Grassland and is classified as endangered, with only 1% conserved [1]. The western boundary of Telperion is the perennial Wilge River [19]. The mean summer rainfall ranges between 650 and 700 mm per annum, with the highest rainfall recorded in January. The average minimum temperature is 7°C and the average maximum is 27°C. Frost occurs in winter from May to August. The average altitude for the reserve is 1350 m.a.s.l.

Three study sites with varying densities of *S. plumosum* were visually selected for this study. All study sites were part of the same broad vegetation type (Rand Highveld Grassland), had similar topography

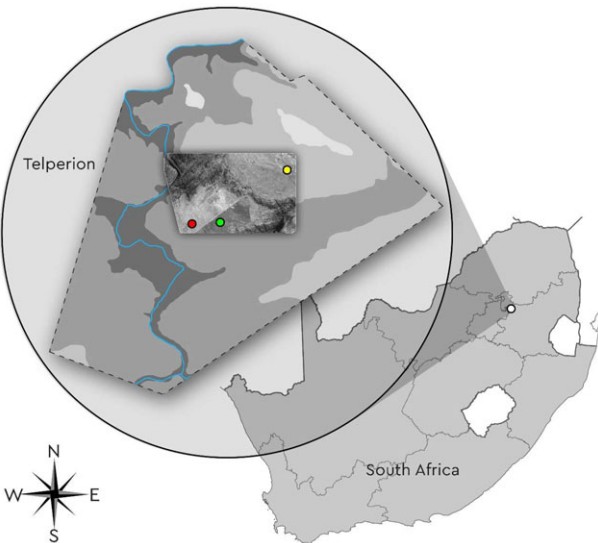

**Figure 1.** Map showing the location of Telperion in the Mpumalanga province of South Africa, with the study sites within Telperion highlighted (red dot, low density; yellow dot, intermediate density; green dot, high density).

and the same broad soil types. Study sites were located between 1 and 3 km of one another, each site was a minimum of 50 m from the road, and all were $100 \times 100$ m in size. One site had almost no *S. plumosum* present (named the Grassland site), one was intermediately densified by *S. plumosum* (named the Intermediate site) and one was severely densified by *S. plumosum* (named the Dense site).

## 2.2. Sampling design

Initial vegetation surveys were conducted in all three sites during February 2017 to calculate the *S. plumosum* frequency and establish the gradient of density between the three study sites. Grass and *S. plumosum* height was measured using a measuring stick at 1 m intervals along a 100 m line transect, the transect was placed diagonally across the study site. At each of the three study sites, all *S. plumosum* plants were counted in two $100 \text{ m}^2$ ($10 \times 10$ m) sample plots. To determine species composition, species richness and species diversity, all plant species within twenty $1 \times 1$ m quadrants were identified and counted within each of the study sites. The biomass of grass vegetation was determined using a disc pasture meter (DPM) that was previously calibrated for the reserve ($n = 100$ readings were collected per sample site) [20]. Since we were interested in the grass component and its grazing value to grassland animal species, only open grass areas in the three sample sites were sampled with the DPM. The effect of *S. plumosum* on herbaceous biomass was not taken into account when calculating the total biomass for the three study sites. All plant species were classified into five successional classes [21], i.e. class 1, pioneer annual; class 2, pioneer perennial; class 3, secondary succession; class 4, secondary succession with anthropogenic disturbance; class 5, climax.

## 2.3. Data analysis

### 2.3.1. Density

Using densities of *S. plumosum* plants in each of the two sample plots at each study site, means were calculated for each study site. Density was reported as the number of plants per hectare.

### 2.3.2. $\alpha$-Diversity

The Shannon–Wiener diversity index [22] was used to calculate $\alpha$-diversity. Index values obtained from diversity index calculations were insufficient for further statistical analysis and were converted into effective numbers [23]. An ANOVA was used to determine whether there were significant differences between the three sites in terms of effective numbers of species (i.e. diversity). A *post hoc* Tukey test was run to determine where the significant difference/s were between the three sites [24].

### 2.3.3. Species evenness

The evenness of species was calculated using the Pielou's evenness index [25].

### 2.3.4. $\beta$-Diversity

The $\beta$-diversity (similarity) for the three sites was interrogated using the extended Sørenson's similarity index [26]. The larger the index value, the greater the similarity. To understand the similarity between the individual sites, the Sorenson's similarity index was calculated for each pair of sites.

### 2.3.5. Vegetation ecological successional classes

A two-way classification $\chi^2$ test was used to determine if there was a significant difference between the three sites based on the observed frequency distribution of the five ecological successional classes.

### 2.3.6. Effect of key ecological variables on vegetation diversity

A constrained correspondence analysis ordination of the three sites (Grassland, Intermediate and Dense) was undertaken using the R software package [27], to determine which variables (effective number, biomass, *S. plumosum* height, grass height and ecological successional status/class) had the greatest effect on the vegetation diversity of the sites.

All statistical analyses were done using the IBM SPSS Statistics v. 25 software package [28]. The chosen critical significance level ($\alpha$) was 0.05.

## 3. Results

### 3.1. Species composition and frequency

The Grassland site had a total of 59 recorded plant species. The five most frequent species and the *S. plumosum* frequency are indicated in figure 2*a*. The Intermediate site had a total of 98 plant species. The five most frequent species and the *S. plumosum* frequency are indicated in figure 2*b*. The five most frequent species as well as the *S. plumosum* frequency for the Dense site are indicated in figure 2*c*. This site had a total of 60 recorded plant species.

The number of shared species (figure 3) between the three sites was 19 and included *Cleome maculata*, *Eragrostis curvula*, *Fimbristylis hispidula*, *Phyllanthus parvulus*, *Vernonia poskeana*, *Perotis patens* and *Melinis repens*.

### 3.2. Species richness and evenness

The intermediate site had the highest species richness of 98 species. The Grassland site had 59 species and the Dense site 60 species—these sites were very similar in terms of species richness. Species richness and evenness for the three sites is depicted using a ranked abundance curve (figure 4), which plots the log abundance data against log species rank order. The graph indicates that overall the Intermediate site lies further right with a more even, flat curve compared with the curves for the Dense or Grassland sites. This indicates that the Intermediate site had a higher species richness and evenness than the other sites.

### 3.3. $\alpha$-Diversity

The $\alpha$-diversity was analysed per site and was converted into effective species numbers (table 1). The Intermediate site was the most diverse (9.72 species m$^{-2}$) with the Grassland site next (7.22 species m$^{-2}$), followed by the Dense site being least diverse (6.32 species m$^{-2}$).

Results from the ANOVA test to compare the effective numbers for 20 plots indicated that there were notable differences between the three sites (one-way ANOVA: $F_{2,57} = 4.92$, $p = 0.01$). Tukey's HSD test results revealed a significant difference between the Grassland and Intermediate sites ($p < 0.00$) and between the Intermediate and Dense sites ($p < 0.00$); however, there was no significant difference between the Grassland and Dense sites ($p = 0.26$).

Sorenson's similarity index values calculated for each pair of sites (table 2), indicated that the lowest similarity occurred between the Intermediate and Dense sites (41%). Similarity between the Grassland and Intermediate sites was also low (42%). The Dense and Grassland sites were most similar (45%).

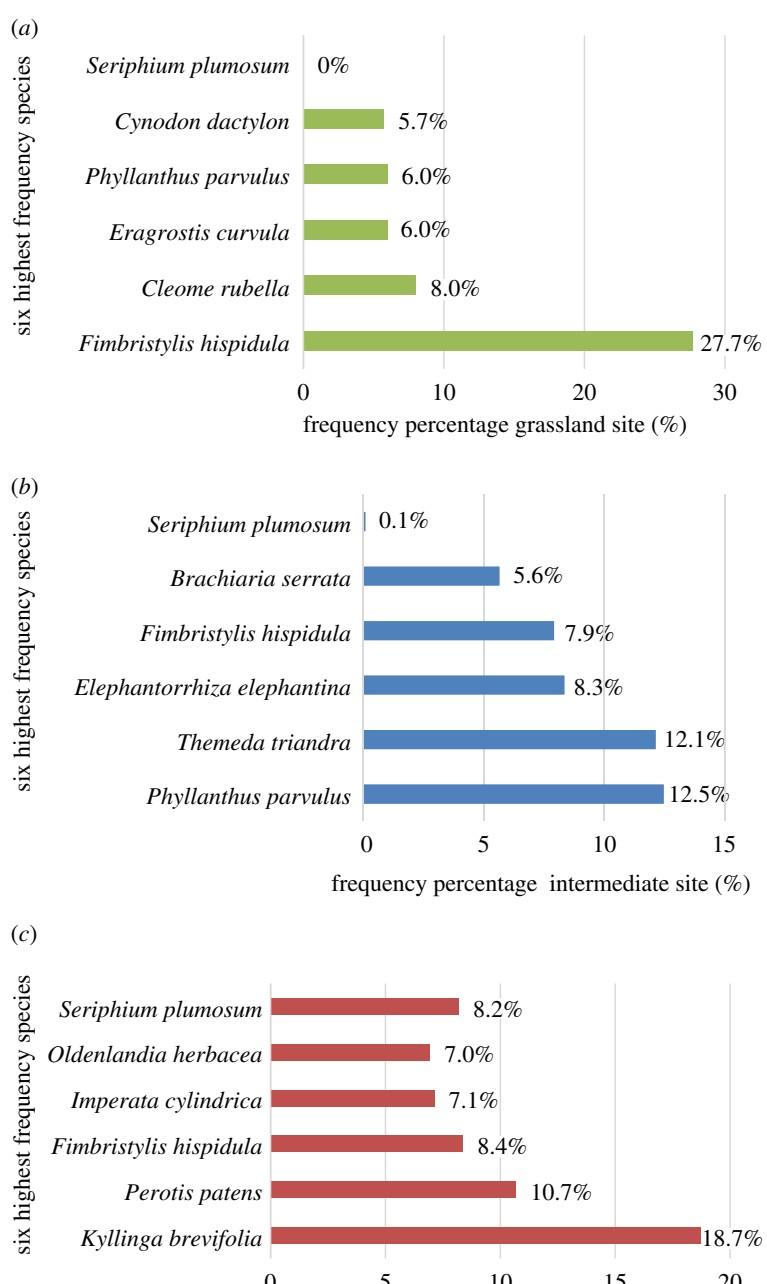

**Figure 2.** Six most frequent species in (*a*) the Grassland site, (*b*) the Intermediate site and (*c*) the Dense site.

These results correlate with the effective numbers from table 1, in that there was no significant difference between the Grassland and Dense sites. When looking at the differences between the sites, *β*-diversity levels for the three sites indicated that they are not similar.

## 3.4. Physiognomy and biomass

*Seriphium plumosum* density in the Grassland site was 24 individuals (ind) ha$^{-1}$, the Intermediate site was 1500 ind ha$^{-1}$ and the Dense site was 9500 ind ha$^{-1}$. Biomass measurements indicated that the Dense site had the highest biomass (9000 kg ha$^{-1}$) followed by the Intermediate (4000 kg ha$^{-1}$) and Grassland (3600 kg ha$^{-1}$) sites, respectively.

The mean grass height was greatest in the Grassland site (594 mm), followed by the Intermediate site (490 mm) and the Dense site (440 mm) (figure 5). *Seriphium plumosum* did not occur along the transect in the Grassland site, therefore no height was recorded. The mean *S. plumosum* height was higher in the

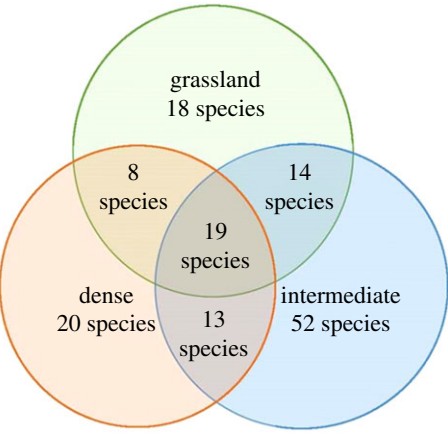

**Figure 3.** Venn diagram indicating the overlap of species at the three study sites within Telperion.

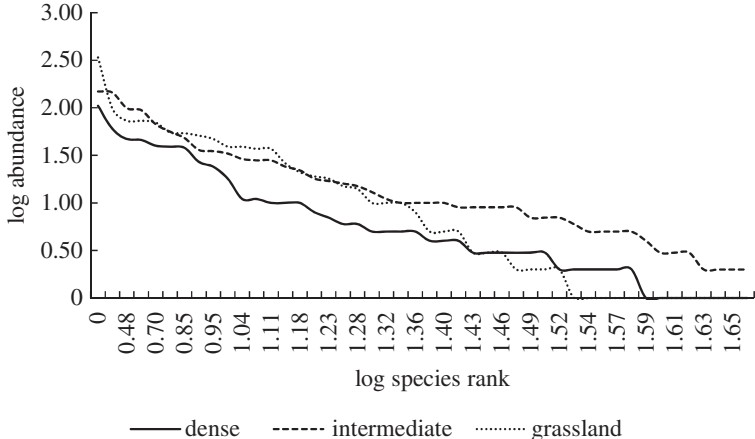

**Figure 4.** Species rank abundance curve for the three sites.

**Table 1.** Species diversity for the three sites, indicating the Shannon–Wiener index, the Simpson's index and the calculated effective numbers based on the average richness per square metre.

| site | species richness (m$^2$) | Shannon–Wiener index | Simpson's index | effective numbers |
|---|---|---|---|---|
| Grassland | 11 | 1.96 | 0.86 | 7.22 |
| Intermediate | 13 | 2.26 | 0.88 | 9.72 |
| Dense | 9 | 1.78 | −0.51 | 6.32 |

Intermediate site (750 mm) than in the Dense site (680 mm). In the Dense site, *S. plumosum* was taller than the grasses, with a resultant greater canopy cover for this species.

## 3.5. Ecological successional status

In terms of ecological successional status (figure 6), the Grassland site was dominated by secondary successional species (52%). Although it had a high percentage of pioneer annuals (30%), it also had the highest percentage of pioneer perennials (14%). There were a few climax species (4%—the lowest of all sites) and no anthropogenic disturbance indicator species. Species composition of the Intermediate site showed a high amount of climax species (42%), fewer secondary successional species (30%) and some anthropogenic disturbance indicator species (9%). This site also had the lowest number of pioneer species (19% perennials and annuals). The Dense site had 17% climax species and was dominated by secondary successional species (47%), it had the second highest percentage of pioneer annual species (18%) and a number of anthropogenic disturbance indicator species (10%).

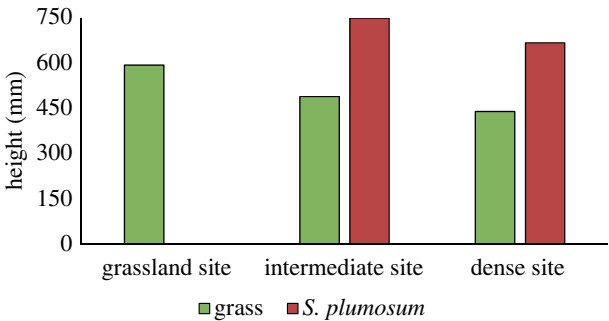

**Figure 5.** Mean grass and *S. plumosum* height (mm) per site. No *S. plumosum* height was recorded in the Grassland site.

**Table 2.** Sorensen similarity index comparing the three sites with one another.

| site | i species count per site A, B and C | ii species count added per site (a + b) (b + c) (a + c) | iii species shared (ab) (bc) (ac) | iv species shared ×2 | Sorensen similarity index iv/ ii (%) | difference (%) |
|---|---|---|---|---|---|---|
| A Grassland and B Intermediate (*ab*) | 59 | 157 | 33 | 66 | 42 | 58 |
| B Intermediate and C Dense (*bc*) | 98 | 158 | 32 | 64 | 41 | 59 |
| A Grassland and C Dense (*ac*) | 60 | 119 | 27 | 54 | 45 | 55 |

Findings indicate that there was a significant difference between the sites in terms of their ecological successional status (two-way classification $\chi^2$: $\chi^2_9 = 729.63$, $N = 3014$, $p = 0.01$).

## 3.6. Ordination

Ordination results (figure 7) indicate that ecological successional status classes (pioneer annual; pioneer perennial; secondary succession; secondary succession with anthropogenic disturbance; climax) and diversity (effective numbers) are strongly associated with the Grassland and the Intermediate sites. There is a strong positive correlation between biomass, *S. plumosum* height and grass height. There is very little correlation between ecological successional status classes and effective numbers; however, ecological successional status was associated with Grassland plots 14 and 16. There is no correlation between effective numbers and *S. plumosum* height.

# 4. Discussion

The Intermediate site is characterized by climax species and is more diverse than the other sites when considering species richness. At Telperion, the Intermediate site represents a grassland with a low density of *S. plumosum* cover (15%), which is more diverse than either the Dense site with a high density (95%) or the Grassland site devoid of *S. plumosum*. This is linked to increased diversity causing ecological stability. As functional richness and diversity increases, plant communities become more resilient [4]. Hobbs & Huenneke [29] state that areas with more microhabitats normally support a higher plant diversity by generating new niches for further 'new' species to establish and flourish.

*Seriphium plumosum* at a high density (9500 plants ha$^{-1}$) negatively affects $\alpha$-diversity in the study area due to allelopathic properties in the plants' organs, which inhibit seed germination and growth of other species [30]. There were notable differences between the three sites in terms of diversity. Tukey's HSD test confirmed that the Intermediate site was significantly more diverse than both the Grassland and Dense sites. When *S. plumosum* is at an intermediate density, results indicate that its

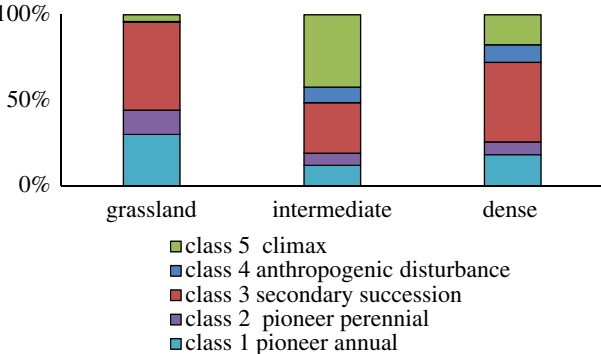

**Figure 6.** Plant species ecological successional status classes (%) for all three sites.

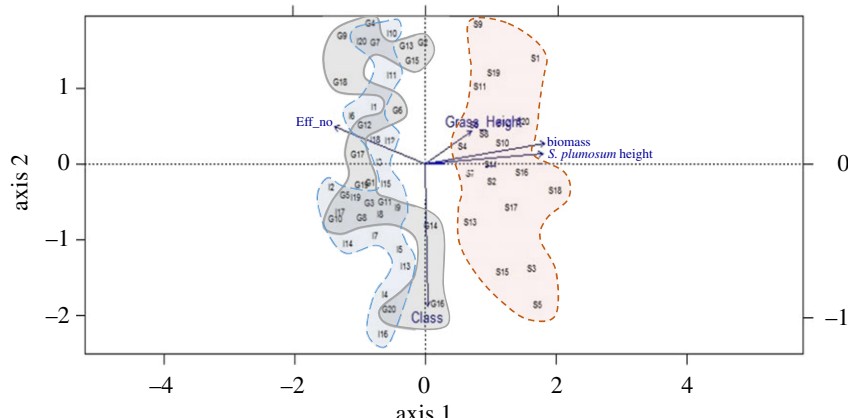

**Figure 7.** Combined ordination diagram for the three sites (grey, Grassland; blue, Intermediate; red, Dense site).

presence leads to an increase in $\alpha$-diversity. This finding is supported by Leps [31], who states that the presence of woody species creates additional niches, encouraging more species to thrive. Similar to the results of this study, Leps [31] states that species diversity tends to decrease on either end of the spectrum. This explains the lower $\alpha$-diversity of the Grassland site caused by the absence of woody species or microhabitats. We speculate that the stable Intermediate site, which is in a climax stage and dominated by the grass species *Themeda triandra*, is able to restrict *S. plumosum* densities and maintain a healthy equilibrium.

Avenant [12] regards high-density sites to have between 10 000 and 20 000 ind ha$^{-1}$. The Dense site in this study is close to these levels of density with 9500 ind ha$^{-1}$. According to Jordaan [14], dense stands of *S. plumosum* (greater than or equal to 10 000 ind ha$^{-1}$) diminish grass growth and primary production by up to 75%. It has also been noted that encroachment is linked to a decrease in ecosystem functions and processes [8–10].

Exclusion of fire and/or heavy grazing leads to encroachment by woody species in grassland and savannah ecosystems [32–34]. Trollope [35] suggests that a lack of controlled burning and hot fires leads to an increase in *S. plumosum* density. High biomass values in the Dense site may be the result of reduced burning in this area. The majority of grass species found in the Dense site are not palatable and, therefore, not used. Soil moisture may differ in the Dense site due to shading by *S. plumosum* which is conducive to the growth of tall grass species; however, this has not been looked at in this study. Another aspect to consider is access to the area by large grazers. Mature *S. plumosum* plants are hard and scratchy, resulting in ungulates avoiding the area. The combination of unpalatable grasses, reduced access and reduced fire may explain the high grass biomass within the Dense site.

There was a significant difference between the three sites in terms of their ecological successional status. The Grassland site was dominated by the perennial grasses, *Eragrostis curvula* and *Cynodon dactylon*. Due to the high percentage of secondary successional species, the Grassland site is regarded as being in a late secondary successional phase. The Intermediate site is dominated by *Themeda triandra* and is considered to be in a climax successional phase as it has a high number of climax

species. By contrast, the Dense site is considered to be degraded as it has been invaded by *S. plumosum* and has a high percentage of pioneer annual species.

The ecological successional status classes (pioneer annual; pioneer perennial; secondary succession; secondary succession with anthropogenic disturbance; climax) and diversity (effective numbers) are strongly associated with the Grassland and Intermediate sites. There is a strong positive correlation between biomass, *S. plumosum* height and grass height.

# 5. Conclusion

Acocks [36] considers Telperion to be a transition zone between the grassland and savannah biomes, both of which are present on the reserve. According to Deacon [37], this is ecologically important and gives rise to the biological diversity present on the reserve. *Seriphium plumosum* density does not have a great impact on α-diversity; however, it does affect plant species composition, structure and ecological successional status of the areas where it is present. At an intermediate level of *S. plumosum* density, α-diversity increases and vegetation structure changes creating microhabitats that allow new species to establish and thrive. Our findings indicate that intermediate densities of *S. plumosum* have positive effects on the ecological successional status and species composition of an area. The presence of *S. plumosum* creates niche areas for other species to establish, which in turn makes for a more resilient community in terms of responding to adverse conditions or changes in the environment. The presence of *S. plumosum* in the Intermediate site results in increased heterogeneity as reflected in the ecological successional status of the site. This is reiterated by Leps [31], who confirms that a community with a higher species richness will have a higher community resilience.

The Grassland site appears to be in a late secondary successional phase, but is in a better ecological condition than the Dense site when comparing biomass, ecological successional status and effective numbers of species present. The Grassland site is a productive, stable environment. The Dense site is impacted by densification of *S. plumosum* and is significantly different to the Intermediate site in terms of the effective number of species, while its ecological successional status and low palatability show that the area is degraded. High biomass values in the Dense site are a concern and may be compounding the problem. *Seriphium plumosum* is highly flammable and combined with high biomass values in the Dense site could cause an intense fire which would be difficult to control. Intense fires sterilize the soil and cause long-term damage to ecosystems. The Dense site is stable in its current condition, which means that for biodiversity to increase in this area, it would require human intervention. At the current high levels of densification in the Dense site, restoration is difficult. To be successful, active interventions would be required, such as physical removal and reseeding of grass species [30]. In the dense site, the dominant *S. plumosum* will continue to outcompete other herbaceous species and the area will remain degraded unless interventions are implemented. Any interventions that are undertaken need to be well planned with regular monitoring and follow-up action to ensure new growth is eradicated.

The Intermediate site is regarded to be representative of a climax grassland due to its dominance by climax plant species and high species diversity. From a grass production perspective, this site has the second highest biomass and better palatability compared with the other two sites. This site is a complex community with a greater variety of plant species, which allows for more variation in species interactions [38].

Biodiversity conservation relies on keeping a balance between permitting and restricting disturbances within ecological thresholds [39]. This study indicates that the presence of *S. plumosum* at low densities in a climax, stable grassland does not have a negative effect on the grassland and to a certain extent even increases the resistance of the community. The conservation of diversity is necessary for maintaining a stable and productive ecosystem [4]. *Seriphium plumosum* at low densities should not be considered a negative aspect of the environment, but a natural part thereof. That said, it is important that areas where *S. plumosum* occurs should be monitored. If this species is no longer in balance within the system and starts to become dense, it will negatively affect biodiversity, species composition and structure of the ecosystem. It is also important to keep this in mind when changes occur in a reserve, with regard to fire and grazing capacity, as these could initiate the densification of this species. All stakeholders, including land managers, decision-makers and researchers need to work together to respond to the causes of biodiversity loss [40].

Ethics. The study uses animals, ethics approvals and permits from MTPA are available under electronic supplementary material. Ethical approval was received from BirdLife SA (2016-05 B) and CAES Animal Research Ethics Committee (2017/CAES/096). Relevant fieldwork details include a permit from Mpumalanga Tourism and Parks Agency (MPB.5568 and MPB.5561) as well as a letter of permission from E Oppenheimer & Son.

Data accessibility. The article's supporting data and materials can be accessed from the Dryad Digital Repository: https://doi.org/10.5061/dryad.2547d7wmt [41]. No code was used.

Authors' contributions. S.C.G. carried out the data collection and fieldwork, carried out the statistical analyses, participated in the design of the study and drafted the manuscript; A.S.B. participated in the design of the study, carried out the statistical analyses and critically revised the manuscript; L.R.B. participated in the design of the study and critically revised the manuscript. All authors gave final approval for publication and agree to be held accountable for the work performed therein.

Competing interests. We declare we have no competing interests

Funding. This work was financially supported by the Oppenheimer Family.

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
