## [Reviewer comments · Royal Society Open Science]

Review History

RSOS-192025.R0 (Original submission)

Review form: Reviewer 1

Is the manuscript scientifically sound in its present form?

Yes

Are the interpretations and conclusions justified by the results?

Yes

Is the language acceptable?

Yes

Do you have any ethical concerns with this paper?

No

Have you any concerns about statistical analyses in this paper?

No

Recommendation?

Accept as is

Comments to the Author(s)

This manuscript provided very valuable information for the practical management of biodiversity at local level. The statistical analyses are relevant and descriptive of the observed biodiversity patterns. It is a well-rounded article and should be published in its current format.

Review form: Reviewer 2**Is the manuscript scientifically sound in its present form?**

Yes

Are the interpretations and conclusions justified by the results?

Yes

Is the language acceptable?

Yes

Do you have any ethical concerns with this paper?

No

Have you any concerns about statistical analyses in this paper?

No

Recommendation?

Accept with minor revision (please list in comments)

Comments to the Author(s)

The results support the hypothesis and this is an important study since chemical treatment degraded vegetation and habitat in which this specific plant species occurs. The Abstract last paragraph is for SA agriculture and nature conservation really important.

Line 9 - encroacher - rather use densifier - which is currently not an English word but should be a new word - otherwise rephrase: the sentence .. example the densification of the indigenous species spreads rapidly..?

line 13 - it is important to quantify the densities - which has been done line 220. The transition between the low to intermediate and high densities should be made clear in abstract and with results.

line 35 - 4 in reference is Diaz?

line 59 - Start sentence with full genus name. same with line 64.

line 96 - Figure 1 - scale and legend?

line 101 - reference for broad plant community

line 105 - encroached rather densified or example above

line 116-150 - references for these methods - international journal - DPM, Tukey test?

line 145 - also for rest of text - first site should be low density, the plant community is the same thus it is all Grassland - therefore it cannot be grassland, intermediate.. - it should be low density - or just low - in line 220 you recorded 24 individuals - which indicate a low density of *S. plumosum* (italics).

line 143 - 5 - write out 5 - five - also elsewhere in text.

line 143 - 5 - write out 5 - five - also elsewhere in text.

line 162 - 169 - Fig. 2 - percentage in b not correct and would suggest it should be not in bar but at the end of the bar. c is barely readable.

line 220 - move the 3.4 paragraph to front - important the quantification.

Line 233 - Figure 5 - Legend? Height

Line 251 - 3 - three

Line 269 – Figure 7 – without colour difficult.
 Line 296 – Avenant number? 12?
 Line 303 – no. 40 & 41 not in references
 Line 349 and line 370 – Seriphium and not S.
 Line 376 – encroachment – should be densification.
 Line 452 – 456 – references not in text.
 Line 456 Stoebe italics.

Decision letter (RSOS-192025.R0)

21-Feb-2020

Dear Mrs Graham

On behalf of the Editors, I am pleased to inform you that your Manuscript RSOS-192025 entitled "Impact of *Seriphium plumosum* densification on Mesic Highveld Grassland biodiversity in South Africa" has been accepted for publication in Royal Society Open Science subject to minor revision in accordance with the referee suggestions. Please find the referees' comments at the end of this email.

The reviewers and handling editors have recommended publication, but also suggest some minor revisions to your manuscript. Therefore, I invite you to respond to the comments and revise your manuscript.

- Ethics statement

- Data accessibility

<http://datadryad.org/submit?journalID=RSOS&manu=RSOS-192025>

- Competing interests

- Authors' contributions

All submissions, other than those with a single author, must include an Authors' Contributions section which individually lists the specific contribution of each author. The list of Authors should meet all of the following criteria; 1) substantial contributions to conception and design, or

acquisition of data, or analysis and interpretation of data; 2) drafting the article or revising it critically for important intellectual content; and 3) final approval of the version to be published.

- Acknowledgements

- Funding statement

Because the schedule for publication is very tight, it is a condition of publication that you submit the revised version of your manuscript before 01-Mar-2020. Please note that the revision deadline will expire at 00.00am on this date. If you do not think you will be able to meet this date please let me know immediately.

- 1) A text file of the manuscript (tex, txt, rtf, docx or doc), references, tables (including captions) and figure captions. Do not upload a PDF as your "Main Document";
- 2) A separate electronic file of each figure (EPS or print-quality PDF preferred (either format should be produced directly from original creation package), or original software format);

- 3) Included a 100 word media summary of your paper when requested at submission. Please ensure you have entered correct contact details (email, institution and telephone) in your user account;
- 4) Included the raw data to support the claims made in your paper. You can either include your data as electronic supplementary material or upload to a repository and include the relevant doi within your manuscript. Make sure it is clear in your data accessibility statement how the data can be accessed;
- 5) All supplementary materials accompanying an accepted article will be treated as in their final form. Note that the Royal Society will neither edit nor typeset supplementary material and it will be hosted as provided. Please ensure that the supplementary material includes the paper details where possible (authors, article title, journal name).

If your manuscript is newly submitted and subsequently accepted for publication, you will be asked to pay the article processing charge, unless you request a waiver and this is approved by Royal Society Publishing. You can find out more about the charges at <https://royalsocietypublishing.org/rsos/charges>. Should you have any queries, please contact openscience@royalsociety.org.

on behalf of Professor Xinguang Zhu (Associate Editor) and Pete Smith (Subject Editor)
openscience@royalsociety.org

Reviewer comments to Author:
Reviewer: 1

Comments to the Author(s)

This manuscript provided very valuable information for the practical management of biodiversity at local level. The statistical analyses are relevant and descriptive of the observed biodiversity patterns. It is a well-rounded article and should be published in its current format.

Reviewer: 2

Comments to the Author(s)

The results support the hypothesis and this is an important study since chemical treatment degraded vegetation and habitat in which this specific plant species occurs. The Abstract last paragraph is for SA agriculture and nature conservation really important.

Line 9 - encroacher - rather use densifier - which is currently not an English word but should be a new word - otherwise rephrase: the sentence .. example the densification of the indigenous species spreads rapidly..?

line 13 - it is important to quantify the densities - which has been done line 220. The transition between the low to intermediate and high densities should be made clear in abstract and with results.

line 35 - 4 in reference is Diaz?

line 59 - Start sentence with full genus name. same with line 64.

line 96 - Figure 1 - scale and legend?

line 101 - reference for broad plant community

line 105 - encroached rather densified or example above

line 116-150 - references for these methods - international journal - DPM, Tukey test?

line 145 - also for rest of text - first site should be low density, the plant community is the same thus it is all Grassland - therefore it cannot be grassland, intermediate.. - it should be low density - or just low - in line 220 you recorded 24 individuals - which indicate a low density of *S. plumosum* (italics).

line 143 - 5 - write out 5 - five - also elsewhere in text.

line 162 - 169 - Fig. 2 - percentage in b not correct and would suggest it should be not in bar but at the end of the bar. c is barely readable.

line 220 - move the 3.4 paragraph to front - important the quantification.

Line 233 - Figure 5 - Legend? Height

Line 251 - 3 - three

Line 269 - Figure 7 - without colour difficult.

Line 296 - Avenant number? 12?

Line 303 - no. 40 & 41 not in references

Line 349 and line 370 - *Seriphium* and not *S.*

Line 376 - encroachment - should be densification.

Line 452 - 456 - references not in text.

Line 456 *Stoebe* italics.

Author's Response to Decision Letter for (RSOS-192025.R0)

See Appendix A.

Decision letter (RSOS-192025.R1)

04-Mar-2020

Dear Mrs Graham,

It is a pleasure to accept your manuscript entitled "Impact of *Seriphium plumosum* densification on Mesic Highveld Grassland biodiversity in South Africa" in its current form for publication in

Royal Society Open Science. The comments of the reviewer(s) who reviewed your manuscript are included at the foot of this letter.

on behalf of Professor Xinguang Zhu (Associate Editor) and Pete Smith (Subject Editor)
openscience@royalsociety.org

Appendix A

Dear Referees	
Thank you for taking the time to review this manuscript, your comments were valuable. We have affected majority of the suggestions, please see attached tracked changes and below table of details.	
Line 9 - encroacher - rather use densifier	Agree, changed
line 13 - it is important to quantify the densities - which has been done line 220.	Agree, changed
line 35 - 4 in reference is Diaz?	Corrected
line 59 - Start sentence with full genus name. same with line 64.	Corrected
Line 96 Figure 1	Legend description is now included in caption. Scale is not relevant in Figure 1 as it only indicates the location of the reserve in South Africa and the general location of the study sites in the reserve.
line 101 - reference for broad plant community	Agree, changed
line 105 - encroached rather densified or example above	Agree, changed
line 116-150 - references for these methods - international journal - DPM, Tukey test?	Corrected
Line 145	Understand the point made but these site names have been defined previously on lines 105 to 107 – all tables and figures contain these names. Thus, this naming convention is correct.
line 143 - 5 - write out 5 - five - also elsewhere in text.	Corrected
Line 162 – 169 Fig 2	Moved percentage to outside the bar. Fig 2 b is correct.
Line 220	The quantification has been stated in the abstract and here we present the results of the physical determination of the density counts for the visually selected sites.
Line 233 – Figure 5 – Legend? Height	Corrected
Line 251 – 3 – three	Corrected
Line 269	Agree that without this in colour it would be difficult to differentiate, therefore it is in colour.
Line 296 – Avenant number? 12?	Corrected
Line 303 – no. 40 & 41 not in references	Corrected
Line 349 and line 370 – Seriphium and not S.	Corrected
Line 376 – encroachment – should be densification.	Corrected

Line 452 – 456 – references not in text.	Corrected
Line 456 Stoebe italics.	Corrected
References	We have used Mendeley, all references have been checked and corrected. If the Mendeley field links are a problem, we can remove them for the final version.